# Electroacupuncture at Fengchi(GB20) and Yanglingquan(GB34) Ameliorates Paralgesia through Microglia-Mediated Neuroinflammation in a Rat Model of Migraine

**DOI:** 10.3390/brainsci13040541

**Published:** 2023-03-24

**Authors:** Min Zhou, Fang Pang, Dongmei Liao, Xinlu He, Yunhao Yang, Chenglin Tang

**Affiliations:** 1College of Traditional Chinese Medicine, Chongqing Medical University, Chongqing 400016, China; 2Chongqing Key Laboratory of Traditional Chinese Medicine for Prevention and Cure of Metabolic Diseases, Chongqing 400016, China

**Keywords:** migraine, electroacupuncture, paralgesia, microglia, inflammatory response

## Abstract

Background: Multiple studies have suggested that paralgesia (hyperalgesia and cutaneous allodynia) in migraine reflects the activation and sensitisation of the trigeminovascular system (TGVS). In particular, it reflects the second-order and higher nerve centre sensitisation, which is caused and maintained by neuroinflammation. Microglia activation leads to the release of proinflammatory cytokines involved in inflammatory responses. Accumulating evidence indicates that electroacupuncture (EA) is effective in ameliorating paralgesia, but the underlying mechanisms of EA in migraine attacks caused by microglia and microglia-mediated inflammatory responses are still unclear. The purpose of this study was to explore whether EA could ameliorate the dysregulation of pain sensation by suppressing microglial activation and the resulting neuroinflammatory response, and to evaluate whether this response was regulated by Toll-like receptor 4 (TLR4)/nuclear factor-kappa B(NF-κB) in the trigeminal nucleus caudalis (TNC) in a rat model of migraine. Methods: Repeated Inflammatory Soup (IS) was infused into the dura for seven sessions to establish a recurrent migraine-like rat model, and EA treatment was administered at Fengchi (GB20) and Yanglingquan (GB34) after daily IS infusion. Facial mechanical withdrawal thresholds were measured to evaluate the change in pain perception, and plasma samples and the TNC tissues of rats were collected to examine the changes in calcitonin gene-related peptide (CGRP), the Ibal-1-labelled microglial activation, and the resulting inflammatory response, including interleukin-1β (IL-1β), tumour necrosis factor-α (TNF-α), interleukin-6 (IL-6), and their regulatory molecules TLR4/NF-κB, via enzyme-linked immunosorbent assay (ELISA), real-time polymerase chain reaction (RT-PCR), immunohistochemistry (IHC) and Western blot analysis. Results: Repeated IS injections into the dura induced facial mechanical paralgesia, which is the manifestation of migraine attacks, and increased the expression of CGRP, Ibal-1, microglial mediated inflammatory cytokines (IL-1β, TNF-α, IL-6), and regulatory molecules TLR4/NF-κB. EA at GB20/34 significantly attenuated repetitive IS-induced pain hypersensitivity. This effect was consistent with decreased levels of CGRP and inflammatory cytokines in the plasma and the TNC via the inhibition of microglia activation, and this response may be regulated by TLR4/NF-κB. Conclusions: EA ameliorated paralgesia in repetitive IS-induced migraine-like rats, which was mainly mediated by a reduction in microglial activation and microglial-mediated inflammatory responses that could be regulated by TLR4/NF-κB.

## 1. Introduction

Migraine is a multifactorial neurological disease characterised by recurrent, moderate, or severe headache [1]. The increasing frequency of headaches is a critical risk factor for the chronification progression of migraine [2]. Hyperalgesia, an over-amplification of the response to painful stimuli, and cutaneous allodynia, a perception of pain in response to normally innocuous stimuli, are usually observed among patients with migraine and represent chronic progression [3]. Dysregulation of pain sensation is a clinical manifestation of migraine attacks, which is attributed to neuroinflammation in the trigemino vascular system (TGVS) [4,5], especially in the trigeminal nucleus caudalis (TNC) [6]. An increasing number of studies show that neuroinflammation plays an important role in the pathophysiological mechanism of migraines and can increase the frequency, and maintain the occurrence, of chronic migraines [6,7]. Unfortunately, there are no specific medications to inhibit and delay the progression of recurrent migraines to chronification processes because the detailed pathomechanism is unclear; hence, research on effective alternative therapies is necessary.

The increase in migraine attack frequency leading to the chronification process relates to the neurogenic inflammatory response and neuroinflammation driven by glia-produced cytokines [8]. Recent evidence has revealed that microglia in the TNC directly or indirectly influence this process [6,7,8,9]. Moreover, numerous studies have shown that microglia and activated microglial-derived cytokines, such as interleukin-1β (IL-1β), tumour necrosis factor-α (TNF-α), and interleukin-6 (IL-6), are involved in pain initiation and the maintenance of chronic pain [10,11]. Moreover, pain hypersensitivity can be significantly alleviated by the inhibition of microglial activation and inflammatory cytokines [6,12]. A previous study demonstrated that the intrathecal injection of IL-1β, TNF-α, or IL-6 elicited pain hypersensitivity in naive animals, confirming that proinflammatory cytokines can directly induce pain hyper sensitivity [13]. In addition, as a surface receptor of microglia, Toll-like receptor4 (TLR4) plays an important role in mediating inflammation signalling pathways by activating the downstream mediator nuclear factor-kappa B (NF-κB), which increases the expression of proinflammatory cytokines [14,15]. These molecules promote the activation of microglia and the release of inflammatory cytokines, resulting in hypersensitivity via the nociceptive pathway [16]. This paralgesia state can be alleviated by TLR4 blockers (e.g., TAK-242), which also inhibit microglial activation and decrease the production of TLR4 downstream molecules and inflammatory cytokines in a rat model of migraine [15]. Accordingly, alleviating neuroinflammation via microglia, microglial-mediated, and correlated inflammatory molecules may be an effective strategy for migraine treatment. Calcitonin gene-related peptide (CGRP) is a neuropeptide that transmits pain information released from trigeminal nerve fibres in the context of inflammation and produces pro-inflammatory mediators [17,18,19]. Previous clinical studies have shown that CGRP could be a biomarker aiding in the diagnosis of chronic migraines because distinctly elevated CGRP levels were observed in plasma samples from patients with chronic migraine, and anti-migraine treatments targeting either CGRP release, CGRP peptide, or its receptor were clinically effective [20].

As a complementary and alternative treatment modality, multiple reports have shown that electroacupuncture (EA) has an analgesic effect through the central pathway and reduces inflammation [21,22]. Multiple studies in EA have shown that it can prevent migraine attacks and ameliorate the pain hypersensitivity of migraines [23,24]. In addition, it has been demonstrated that EA can show anti-migraine effects in experimental migraine rat models [25,26]. Some studies have gradually focused on the role of EA in the central pathway in recent years. However, there are concerns about the inhibition of hypersensitivity by derangement of monoamine neurotransmitters [25,27,28]. Generally, GB20 (Fengchi) is a high-frequency acupoint used in migraine research, but it is always used in combination with multiple acupoints for clinical migraine treatment [24,25,27]. Most importantly, compared to GB20 alone, many previous studies have already confirmed that EA at GB20 and GB34 could maximise the effects [28,29,30].

Based on the aforementioned evidence, we hypothesised that EA treatment at GB20/GB34 would alleviate pain hypersensitivity in a rat model of migraine by ameliorating neuroinflammation via the inhibition of microglia and a decrease in microglial-mediated inflammatory cytokines (IL-1β, TNF-α and IL-6), and that microglial activation and relevant inflammatory responses might be regulated by microglial TLR4 and the downstream molecule NF-κB. To test this hypothesis, we mimicked the migraine-like rat model previously used to induce recurrent migraine attacks and hyperalgesia following repeated inflammatory soup (IS) injections into the dura [31]. We then evaluated mechanical allodynia using von Frey filaments and analysed CGRP expression in plasma and the TNC by enzyme-linked immunosorbent assay (ELISA) and real-time polymerase chain reaction (RT-PCR). The same method was used to detect the release of inflammatory cytokines including IL-1β, TNF-α, and IL-6. Immunohistochemistry (IHC) was used to observe the positive expression of microglia, the above inflammatory cytokines, and the morphologic changes of microglia. A Western blot assay was used to detect the protein expression of microglial markers Ibal-1, TLR4, and NF-κB in the TNC.

## 2. Materials and Methods

### 2.1. Animals

Male adult Sprague Dawley rats weighing 200–250 g (Experimental Animal Centre of Chongqing Medical University, Chongqing, China, Certificate number: SYXK (Yu) 2022-0010) were housed in a temperature—(20–25 °C) and humidity—(50–70%) controlled room with ad libitum access to water and food, under a 12 h light/dark cycle for at least 1 week before any experimental procedures were initiated. All experimental procedures involving the use of animals in this study were approved by the Ethics Committee of Chongqing Medical University in China. Animal experiments were also carried out according to the ethical guidelines of the International Association for the Study of Pain [32], and every effort was made to minimise animal suffering.

### 2.2. Experimental Design

#### 2.2.1. Experiment I

To determine the therapeutic effect of EA, we established a recurrent migraine-like rat model via implantation of dural cannula and repetitive IS injections, and randomly established five groups (*n* = 10 per group) for the different corresponding interventions, as well as for conducting sensory sensitivity testing seven days after cannula implantation and surgery recovery (day 0). Subsequent time points (day 1, day 3, day 5, and day 7), and post-treatment assessments including ELISA and RT-PCR were performed to measure the expression of CGRP in plasma and the TNC, with the aim to investigate whether repeated IS injections caused hyperalgesia in migraines, which was consistent with the changes in CGRP, and to investigate the anti-hyperalgesia effect of EA in migraine-like rats.

#### 2.2.2. Experiment II

The morphological changes and positive expression of microglia labelled with ionised calcium binding adapter molecule-1 (Ibal-1) in the TNC, and the expression of inflammatory cytokines, including IL-1β, TNF-α, and IL-6 in plasma and the TNC, were examined using ELISA, RT-PCR, and IHC. This was conducted to investigate whether repeated IS infusions lead to neurogenic inflammation caused by microglial activation, resulting in the release of inflammatory cytokines, and whether EA ameliorates hyperalgesia by reducing microglial activation and relevant inflammatory responses.

#### 2.2.3. Experiment III

To evaluate whether microglial activation and microglial-mediated inflammatory response are regulated by microglial TLR4 and its downstream molecule NF-κB, and whether EA could affect the activation of microglia and the relevant inflammatory response by modulating TLR4/NF-κB, the expression of Ibal-1, TLR4, and NF-κB in the TNC were analysed using Western blot analysis.

#### 2.2.4. Implantation and Fixation of the Cannula

The surgical protocols were performed as described previously [31,33]. All rats, except those in the blank group, were implanted with a cannula after an acclimation period of seven days to perform daily dural infusions. After rats were fasted without food and water for 12 h to prevent abdominal dilation, they were generally anaesthetised with 3% pentobarbital sodium (3 mL/kg, i.p.) and positioned in a stereotaxic apparatus. A longitudinal incision was made and the connective tissue was removed to expose the skull. We drilled through the skull to make a 1.0 mm diameter cranial hole (1.0 mm posterior to the bregma, and 1.5 mm lateral to midline) carefully, and then inserted a stainless steel inner cannula (Catalogue No. 62001, RWD Life Science, Shenzhen, China) into the cranial hole to deliver IS or phosphate-buffered saline (PBS). The cannula was surrounded by dental cement to secure it. Finally, a matching cap (Catalogue No. 62101, RWD Life Science, Shenzhen, China) was inserted into the cannula for sealing, which remained outside the skin after the skin was sutured. It was important that injury of the dura mater be avoided. Post-cannulation, rats were returned to their separate cages and allowed to recover for seven days. Their condition was observed and the surgical wounds were disinfected daily during this period. The seventh day of post-surgery recovery was recorded as day 0 of the following experiments.

#### 2.2.5. Group Assignment and Repeated Injections of Is or PBS into Dura

After a recovery period of 7 days, 50 rats were randomly assigned to the following 5 experimental groups (*n* = 10 for each group): (a) a blank group (Blank), which was fed normally and only participated in the experimental process of grasping and fixation; (b) a PBS group (PBS), which received repeated daily infusion of 10 µL PBS via cannula into the dura; (c) an IS model group (IS), which received repeated daily injection of 10 µL IS through cannula onto the dura; (d) a sham-electroacupuncture group (SEA), in which acupuncture needles were inserted into distant sham-acupoints (approximately 10 mm above the iliac crest) and were only connected to the stimulator without current stimulation after daily 10 µL IS injection; and (e) an electroacupuncture group (EA), which received EA at GB20 and GB34 after daily 10 µL IS injection. The IS was a mixture of inflammatory mediators, consisting of 1 mM histamine (H7375, Sigma-Alrich, St. Louis, MO, USA), 1 mM serotonin (H9523, Sigma-Alrich, St. Louis, MO, USA), 1 mM bradykinin (A304378, Aladdin, Shanghai, China), and 0.1 mM prostaglandin (P5640, Sigma-Alrich, St. Louis, MO, USA) mixed in PBS at pH 7.4 [34,35], and was microinjected through a tube into the epidural cannula. The infusion procedure was performed as described in previous studies [31,36]. A micro-infusion pump was connected to the top of the cannula by a polyethylene tube (Catalogue No. 62329, RWD Life Science, Shenzhen, China), allowing for a steady infusion of 10 µL of IS or PBS over 5 min at a rate of 2 µL/min and the rats were freely moving during this period. The same procedure was repeated once daily for seven days.

### 2.3. Electroacupuncture

According to previous studies [28,30], GB20 and GB34 were located at 3 mm lateral to the midpoint of a line joining the two ears at the back of the head, and the depression anterior and distal to the head of the fibula, respectively, and sham-acupoints were selected at approximately 10 mm above the iliac crest [28,30]. First, rat movement was restricted with the rat-fixating device to adequately expose their heads and legs for intervention, and target points were sterilised with 75% alcohol in the EA and SEA groups. Stainless-steel acupuncture needles (13 mm × 0.25 mm; Suzhou Medical Appliance Factory, Suzhou, Jiangsu Province, China) were inserted at GB20 and GB34 bilaterally to a depth of 2–3 mm, and needle handles were connected to an electrical stimulator (SDZ-II, Suzhou Medical Appliance Factory, Suzhou, Jiangsu Province, China), which was set at a frequency of 2/15 Hz (amplitude-modulated wave) and an intensity of 0.5–1.0 mA for 20 min/day in the EA group. For rats in the SEA group, needles were inserted bilaterally at sham-acupoints to a depth of 2–3 mm and needle handles were connected to an electrical stimulator without current stimulation. Although acupuncture and electric stimulation were not performed in other groups, the rats in these groups were still placed into the fixtures like those in the EA and SEA groups. All rats remained conscious during the treatment and the same process was repeated once daily for seven days.

### 2.4. Tactile Sensory Testing

To test nociceptive sensitivity, von Frey filaments (Aesthesio, Danmic Global, San Jose, CA, USA) with different gram weights and certain stiffnesses were applied via the up-and-down method [37,38], which means that a heavier filament (up) was used in the absence of a response and a lighter filament (down) was tested in the presence of a response. After rats adapted to the test environment for at least 30 min, they were individually placed in transparent perspex chambers, the top of which could be opened to allow filaments to fully touch their face. The filament was applied to the periorbital area with steady vertical pressure to cause slight flexion until a positive withdrawal response was observed. A positive facial withdrawal response [39,40] was defined as quickly retracting the head from the stimulus, stroking the face with the ipsilateral forepaw, and vocalising. The nociceptive threshold was defined as the filament weight at which rats exhibited a positive response with at least three out of five applications. Each filament was tested at least three times with an interval of at least one minute and the gram weight corresponding to the positive reaction was recorded as the mechanical withdrawal threshold. Based on a previous study [36], a 26 g filament was selected as the maximum limit for testing; thus rats that did not respond to the 26 g stimulus were assigned 26 g as their mechanical withdrawal threshold for analysis. All rats were measured for basal threshold on day 0 as the baseline and then once every other day until the end of the experiment (day 7) in an awake and freely moving state to evaluate changes in nociceptive sensitivity. All testing was performed blind to the group assignment.

### 2.5. Tissue Preparation

At the end of all the experiments, the rats were anesthetised with 3% pentobarbital sodium (3 mL/kg, i.p.) and the blood samples (4 mL/per rat) were collected from the abdominal aorta. The samples were then mixed with ethylenediaminetetraacetic acid (EDTA) in anticoagulant tubes and centrifuged at 4000 rpm for 20 min at 4 °C. The supernatant was separated and stored at −80 °C until ELISA analysis. After blood sample collections, four rats in each group were randomly selected to expose their hearts for transcardial perfusion with 0.9% normal saline at 37 °C followed by cold 4% paraformaldehyde (PFA). They were euthanised to harvest brains and the TNC tissues were excised into vials containing 4% PFA solution to fix overnight for the subsequent paraffin-section and IHC analysis. The remaining rats were immediately euthanised after blood sample withdrawal and the TNC tissues were dissected and frozen at −80 °C for subsequent RT-PCR and Western blot experiments. All the operations to separate the brain tissue were performed on ice. A diagram of the experimental protocol is shown in Figure 1.

### 2.6. ELISA Analysis

The concentrations of CGRP and inflammatory cytokines, including IL-1β, TNF-α, and IL-6 in plasma, were determined using rat-specific ELISA kits (MEIKE, Zhangzhou, China) according to the manufacturer’s instructions, and the samples were added to the corresponding micro-ElISA strip-plate wells. Specific HRP-conjugated antibodies were added to each sample for incubation, followed by chromogen solutions A and B for colouring, and, finally, stop solution was added to terminate the reaction with visible colour from blue to yellow. The optical density (OD) was calculated using a microplate spectrophotometer at a wavelength of 450 nm. Each sample was analysed in duplicate, and the mean concentrations were calculated and expressed as picograms of antigens per gram of protein.

### 2.7. RT-PCR Analysis

CGRP and inflammatory cytokines, including IL-1β, TNF-α, and IL-6 mRNA expressions in the TNC, were analysed via RT-PCR analysis, as previously described [7,41]. Total RNA was extracted using an RNAiso plus reagent (9108, Takara, Kusatsu City, Japan), and purity and concentration were measured spectrophotometrically using a NanoDrop spectrophotometer (Thermo Fisher Scientific, Waltham, MA, USA). Next, cDNAs were synthesised at 37 °C for 15 min, 85 °C for 5 s, and 4 °C until the end, using the PrimeScript™ RT Reagent Kit with gDNA Eraser (RR047A, Takara, Kusatsu City, Japan) and the T100™ thermoCycler (Bio-Rad, Hercules, MA, USA). Moreover, RT-PCR was performed with TB Green^®^ Premix Ex Taq™ II (RR820, Takara, Kusatsu City, Japan) using a CFX96 Touch thermocycler (Bio-Rad, Hercules, MA, USA) for amplification. PCR cycle conditions were as follows: first, predenaturation at 95 °C for 2 min, followed by denaturation at 95 °C for 5 s, and annealing at 60 °C for 30 s for 40 cycles. All fluorescence data were processed by a post-PCR data analysis software program, and the 2^−ΔΔCT^ relative quantitative method was used to calculate the relative abundance of each target gene mRNA. β-actin mRNA was used as an endogenous control for normalisation. Gene-specific primers were obtained from Sangon Biotech (Shanghai, China). The sequence-specific primers for CGRP and inflammatory cytokines, including IL-1β, TNF-α, IL-6, and β-actin, are shown in Table 1.

### 2.8. Immunohistochemistry (IHC) Staining

IHC staining was used to detect the morphologic changes and positive expression of microglia labelled with Ibal-1, and the resulting positive expressions of inflammatory cytokines. Sections were incubated in 3% H_2_O_2_ for 15 min after routine dewaxing and hydration and then washed 2 times with PBS. The sections were then placed in citrate buffer for antigen repair at high-temperature, and then, after being naturally cooled to room temperature, blocked with 3%normal goat serum for 1 h. This was followed by incubation with each primary antibody overnight at 4 °C: rabbit anti-ibal-1, rabbit anti-IL-1β, rabbit anti-TNF-α, and rabbit anti-IL-6. The next day, the sections were washed 3 times with PBS containing 0.3% Triton X-100 (PBST), incubated with goat anti-rabbit secondary antibodies for 30 min, and washed 5 times with PBST. Diaminobenzidine (DAB) was used to visualise staining, and the sections were observed under a biological microscope after counterstaining with haematoxylin, and being dehydrated, cleared, sealed in turn, and stained with brownish yellow or tan particles. The percentage of positively expressed regions in the total observed regions was examined and analysed using Image J. Specific antibodies and their dilution ratios are listed in Table 2.

### 2.9. Western Blot Analysis

To confirm the microglial activation, and the microglia’s surface receptor and the related signalling pathway involved, the protein expressions of Ibal-1, TLR4, and theTLR4’s downstream molecule, NF-κB p65, in the TNC tissues were measured by Western blot. Firstly, the TNC samples were homogenised in RIPA Lysis Buffer and phenylmethanesulfonyl fluoride at a ratio of 100:1, and then centrifuged (4 °C, 12,000 rpm, 20 min, Allsheng, Hangzhou, China) and extracted for total protein. The protein concentration was quantified according to the BCA protein assay method with a BCA protein assay kit (Beyotime, Shanghai, China). Subsequently, equal amounts of protein specimens were separated on a sodium dodecyl sulfate polyacrylamide gel electrophoresis (SDS-PAGE) gel, and transferred to polyvinylidene difluoride (PVDF) membranes. Membranes were blocked in 5% non-fat milk for 2 h at room temperature, washed three times with tris-buffered saline (TBS) containing 0.1% Tween 20 (TBST), and incubated overnight at 4 °C with each of the primary antibodies (rabbit anti-Ibal-1, mouse anti-TLR4, mouse anti-NF-κB p65, and mouse anti-β-actin) which were diluted in primary antibody dilution buffer (Beyotime, Shanghai, China) in advance. The next day, the protein bands were washed three times with TBST and incubated with the appropriate HRP-conjugated secondary antibodies, goat anti-rabbit IgG or goat anti-mouse IgG, for 2 h at room temperature. Finally, protein bands were visualised using the ECL hypersensitive luminescence kit (4A Biotech, China), and the digital images were acquired using an automatic exposure instrument. The ImageJ analysis system (Fusion, Germany) was used to quantify the optical density values of the detected proteins and β-actin was used as a loading control to normalise protein levels. Specific antibodies and their dilution ratios are listed in Table 2.

### 2.10. Statistical Analysis

Statistical analysis was performed with GraphPad Prism version 8.0 and all data were presented as mean ± standard deviation. Data were analysed using one-way analysis of variance (ANOVA) and post hoc multiple comparisons were performed using Tukey’s test. Kruskal–Wallis test was used to analyse multiple distributed data sets followed by Dunn’s multiple comparisons. Values of *p* < 0.05 were considered statistically significant.

## 3. Results

### 3.1. EA Inhibits the Reduction of Facial Mechanical Withdrawal Thresholds in the Repeated IS-Induced Recurrent Migraine-like Rat Model

To evaluate the changes in the mechanical hyperalgesia following daily injection with IS into the dura and the anti-hyperalgesia effect of EA, facial mechanical withdrawal thresholds were measured on day 0 and at different time points (day 1, day 3, day 5, and day 7) (*n* = 8/group). They were not significantly different from facial withdrawal thresholds in each group before injection with IS (*p* > 0.05) or PBS on day 0 (*p* > 0.05) or after the first injection and corresponding intervention on day 1 (*p* > 0.05). However, repeated IS stimulation decreased the facial withdrawal threshold on day 3 compared with the Blank group (*p* < 0.01) and PBS group (*p* < 0.01), whereas the withdrawal threshold was significantly higher in the EA group than in the IS group (*p* < 0.05) on the same day. In addition, the Blank group did not differ significantly compared to the PBS group (*p* > 0.05) and the SEA group did not differ distinctly from the IS group (*p* > 0.05). Moreover, IS significantly reduced the facial withdrawal threshold compared to the Blank group and the PBS group on day 5 (Blank vs. IS *p* < 0.01; PBS vs. IS *p* < 0.001) and day 7 (Blank vs. IS *p* < 0.001; PBS vs. IS *p* < 0.01). In contrast, compared to IS stimulation alone, EA consistently alleviated hyperalgesia on day 5 (*p* < 0.05) and day 7 (*p* < 0.05). These data indicate that the rats developed mechanical hyperalgesia after repeated IS injections into the dura, while EA relieved these hyperalgesia effects. In contrast, there was no obvious difference between the Blank and PBS groups (*p* > 0.05), which indicated that rats injected with PBS showed no change in sensory threshold and did not establish mechanical allodynia. Furthermore, the SEA group did not significantly differ in the mechanical withdrawal threshold compared to the IS group (*p* > 0.05). As suggested, the sham-acupoints without current stimulation did not exert anti-hyperalgesia effects. A curve graph shows the changes in facial mechanical withdrawal threshold at five time points.

### 3.2. EA Reduces the Concentrations of the CGRP and Inflammatory Cytokines in Plasma

Based on the above sensory testing results and previous research [40,42], repeated IS stimulation reliably induced significant pain hypersensitivity. To investigate whether this hyperalgesia evoked by repeated IS infusions corresponded to increased CGRP and inflammatory cytokine expressions, concentrations of CGRP and inflammatory cytokines, including IL-1β, TNF-α, and IL-6, were analysed after repeated IS injections. More importantly, to determine whether the improvement of pain hypersensitivity by EA corresponded to the inhibition of CGRP and the release of inflammatory cytokines, the relevant levels in each group were examined by ELISA. As shown in Figure 2, the data revealed that the IS group significantly elevated the plasma levels of CGRP, IL-1β, TNF-α, and IL-6 relative to those observed in the Blank group (CGRP: *p* < 0.001; IL-1β: *p* < 0.001; TNF-α: *p* < 0.0001; IL-6: *p* < 0.0001) and the PBS group (CGRP: *p* < 0.05; IL-1β: *p* < 0.001; TNF-α: *p* < 0.001; IL-6: *p* < 0.0001). When compared with the levels observed in the IS group, the plasma levels of CGRP and inflammatory cytokines in the EA group (CGRP: *p* < 0.05; IL-1β: *p* < 0.05; TNF-α: *p* < 0.05; IL-6: *p* < 0.01) were obviously reduced, whereas no significant differences in CGRP and inflammatory cytokine levels were observed between the IS and SEA groups (all: *p* > 0.05). In addition, the levels of CGRP and IL-1β, TNF-α, and IL-6 in plasma did not distinctly differ in the Blank and PBS groups (all: *p* > 0.05); accordingly, repeated IS injections could trigger a large release of CGRP and the inflammatory cytokines in plasma, which was consistent with the increased hyperalgesia, whereas EA markedly reduced the release of CGRP and the inflammatory cytokines, and SEA treatment had no significant effect.

### 3.3. EA Depressed the Releases of CGRP and Inflammatory Cytokines in the TNC

To investigate whether repeated IS injections caused increases in both CGRP and inflammatory cytokines in the TNC, the above indices were further evaluated in the TNC by RT-PCR analysis (*n* = 3–4/group). As shown in Figure 3, it was observed that the mRNA expression of CGRP and inflammatory cytokines, including IL-1β, TNF-α, and IL-6, were evidently higher in the IS group than in the Blank group (CGRP: *p* < 0.01; IL-1β: *p* < 0.001; TNF-α: *p* < 0.05; IL-6: *p* < 0.001) and PBS group (CGRP: *p* < 0.01; IL-1β: *p* < 0.001; TNF-α: *p* < 0.01; IL-6: *p* < 0.001), and that the EA group exhibited distinctly lower mRNA levels of CGRP and inflammatory cytokines than the IS group (CGRP: *p* < 0.05; IL-1β: *p* < 0.01; TNF-α: *p* < 0.05; IL-6: *p* < 0.05). Similarly, there were no obvious differences between the Blank and the PBS group (all: *p* > 0.05), and the SEA group did not differ significantly compared to those in the IS group (all: *p* > 0.05). Based on the changes in the facial mechanical withdrawal threshold, and the results of CGRP both in the Plasma and the TNC in migraine rats, the hyperalgesia of the recurrent migraine-like model induced by repeated IS stimulation was clear, suggesting that repeated injections of IS into the dura could lead to the elevation of CGRP and inflammatory cytokines in the TNC, which is an important site of hyperalgesia resulting from the chronic progression of a recurrent migraine, and that this effect can be negatively regulated via EA.

### 3.4. EA Attenuates Microglial Activation and Inflammatory Cytokines Release in the TNC

To study the effect of repeated IS injections on microglial activation in the TNC, and whether the subsequent release of inflammatory cytokines was consistent with the activation of microglia and, more importantly, the role of EA treatment, the positive expression of Ibal-1-labelled microglia and inflammatory cytokines in the TNC of each group were detected by an IHC assay.

#### IHC Assay for Ibal-1 Labelled Microglia

The study assessed whether repeated IS infusions induced microglial activation in the TNC accordingto morphology or quantity, and whether EA could reduce the activation of microglia (*n* = 4/group). Microglia in the TNC were immunolabelled with Ibal-1. Representative microphotographs and results depicting Ibal-1 immunopositive cells in the TNC are shown in Figure 4. As shown in the Figure, the morphological changes in microglia mainly displayed an activated phenotype, such as hypertrophy, and the positive cell expression of microglia labelled with Ibal-1 in the IS group, after repeated IS injections, had an obvious increase compared with the Blank group (*p* < 0.0001) and the PBS group (*p* < 0.0001), while the amount and staining intensity, as well as the activated phenotype, of microglia in the EA group were decreased compared to the IS group (*p* < 0.001). Moreover, there were no significant changes between the Blank and PBS groups (*p* > 0.05) or between the IS and SEA groups (*p* > 0.05) for microglial phenotype and quantity in the TNC. This indicated that repeated IS injections could cause a transformation of microglia from the resting state to the activated state in the TNC. There was also an increase in the number of microglia, which supported the notion that the sustained activation of microglia was related to repeated IS induction. It is worth mentioning that this effect can be ameliorated by EA to a certain extent.

### 3.5. IHC Assay for Inflammatory Cytokines

The quantities of IL-1β, TNF-α, and IL-6 in each group were further tested in the TNC and similar results with ELISA and PCR were obtained via IHC analysis (*n* = 4/group), as shown in Figure 5, compared with the Blank group (IL-1β: *p* < 0.0001; TNF-α: *p* < 0.0001; IL-6: *p* < 0.0001) and the PBS group (IL-1β: *p* < 0.0001; TNF-α: *p* < 0.0001; IL-6: *p* < 0.0001), which were characterised by a low number of positive cells. The positive cell expressions of the above inflammatory cytokines in the IS group increased. This increase featured an increased number of positive cell and brownish yellow or tan particle staining, but no obvious difference was detected between the Blank group and the PBS group (all: *p* > 0.05). Furthermore, EA intervention significantly reduced the expression of inflammatory cytokines compared to the IS group (IL-1β: *p* < 0.0001; TNF-α: *p* < 0.001; IL-6: *p* < 0.001), while the SEA group did not differ from the IS group (all: *p* > 0.05). These results once again demonstrate that an inflammatory response occurred in the TNC with IS-induced recurrent migraine, which was reduced after EA treatment.

### 3.6. EA Decreases the Protein Expressions of Microglia, Microglial TLR4, and the Downstream Molecule NF-κB in the TNC

It was speculated that microglial activation-induced nociceptive abnormalities were related to microglias’ surface receptors; therefore, to further investigate microglial activation and related possible mechanisms, Western blotting was used to detect the protein expression of Ibal-1, TLR4, and the downstream molecule NF-κB (*n* = 3/group). As shown in Figure 6, the results of microglial protein expressions were similar to those obtained by IHC. Repetitive IS-induced rats expressed significantly more Ibal-1 in the TNC compared to the Blank group (*p* < 0.05) and the PBS group (*p* < 0.05), while the EA group had significantly lower Ibal-1 levels compared to the IS group (*p* < 0.05). As expected, the levels in the Blank and PBS groups (*p* > 0.05) and the SEA and IS groups were not significantly different (*p* > 0.05). As a surface receptor on microglia, TLR4 played a key role in microglia activation [43] and was observed on more protein expressions via repeated IS injections than the Blank group (*p* < 0.01) and the PBS group (*p* < 0.01), whereas there was lower protein expression in the EA group (*p* < 0.01). NF-κB, a significant factor in the downstream signalling pathway mediated by TLR4, was also abundantly expressed in rats in the IS group compared to the Blank group (*p* < 0.05) and the PBS group (*p* < 0.05), while the levels were remarkably reduced in the EA group compared to the IS group (*p* < 0.01). Moreover, the protein expressions of both TLR4 and NF-κB in the Blank and PBS groups (*p* > 0.05) and the SEA and IS groups (*p* > 0.05) did not differ significantly. The mechanism of IS-induced recurrent migraine is mainly neurogenic inflammation triggered by microglial activation, which may be related to the surface receptor and the release of the downstream factor NF-κB, both of which could regulate microglial activation to involve the microglial-mediated inflammatory response. Notably, the protein expression of Ibal-1, TLR4, and NF-κB could be significantly reduced by EA treatment.

## 4. Discussion

The results from this study showed that repeated IS injections can induce mechanical nociceptive abnormalities and an increase in CGRP release. These observed behavioural changes were consistent with the neurochemical changes in CGRP involvement. Moreover, in the TNCs of these repetitive dural IS stimulation rats, it was found that microglial activation and a significant release of microglia-mediated inflammatory cytokines (IL-1β, TNF-α and IL-6), and this phenomena might be regulated by TLR4 and NF-κB. Furthermore, it was observed that the above inflammatory factors and CGRP also increased in the plasma of migraine rats. EA at GB20/34 ameliorated hyperalgesia, the expression of migraine biomarkers (CGRP), the activation of microglia, and factors associated with inflammation caused by repeated IS-induced migraine. These data provide evidence of the key role of microglial activation and factors associated with neuroinflammation in recurrent migraine, and suggest that EA can attenuate this condition by regulating microglial activation and neuroinflammation in the TNC, in addition to systemic inflammation.

### 4.1. EA Can Ameliorate Mechanical Paralgesia and CGRP Releases Following Repeated Dural Stimulation with IS

Repeated dura mater stimulation with IS can directly activate TGVS, neurogenic inflammation, and neuronal sensitisation to produce a chronic state of TGVS activation [34,40,44]. Previous studies have demonstrated that repeated dural inflammatory stimulation can cause cutaneous allodynia and trigeminal sensitisation in rats [45], and cutaneous allodynia is thought to be the characteristic manifestation of migraine chronification, which mainly occurs in the TNC and higher central neurons [7,46]. In this study, the cutaneous receptive field pain threshold significantly decreased following the third IS infusion and reached a low level after the seventh IS infusion, while the PBS infusion failed, suggesting that the induction of hyperalgesia was IS-stimulated, and not stimulated by PBS, and that repeated administration of IS produced stronger hyperalgesia and the progression of chronification. Moreover, these results were consistent with previous investigations on the effects of IS on cutaneous hyperalgesia, which is a consequence of the hyper-responsiveness of nociceptive neurons in the TGVS to repeated stimulation [28,40]. It is worth mentioning that the mechanical withdrawal thresholds did not change significantly after the first IS injection. It was speculated that chemical stimulation did not induce cutaneous allodynia because it is mostly observed in frequent or chronic migraines [47]. GB20 and GB34 are commonly used acupoints to alleviate pain in migraine both in clinical practice and in research [24], and data have shown that EA exerts anti-hyperalgesia effects in a rat model of migraines [27,28,48]. In agreement with these studies, this study demonstrated that EA at GB20/34 significantly alleviated allodynia and improved the mechanical withdrawal threshold, whereas acupuncture at sham-acupoints without current stimulation failed, indicating that the improvement of these behavioural measures was specific to EA treatment.

CGRP, a key neuropeptide in the trigeminal system, is considered an endogenous migraine generator, which plays an important role in the occurrence and maintenance of migraines [49], leads to the initiation of migraines by releasing multiple inflammatory responses to aggravate neurogenic inflammation [50], causes sensitisation of second-order neurones, and promotes the development of migraine chronification [51,52]. Numerous reliable studies suggest that the release of migraine biomarkers, such as CGRP and c-Fos, in the TNC can improve hyperalgesia related to chronic migraines [6,7,45]. Furthermore, previous clinical research has found changes in CGRP concentrations of blood in migraine patients [20]. In the present study, the levels of CGRP from plasma and the TNC were evaluated, and the results suggested that CGRP expression significantly increased in either plasma or the TNC after repeated IS stimulation, while the CGRP release decreased after EA intervention, which is consistent with multiple previous studies [29,51], and demonstrated that the pain hypersensitivity induced by repeated IS stimulation corresponded to an increase in CGRP expression. More importantly, EA could be involved in the pathological process of migraine by reducing CGRP release, resulting in the improvement of pain hypersensitivity. Previous research has identified that increased blood–brain barrier (BBB) permeability in the TNC occurs during the onset and chronification of trigeminal pain following dural stimulation with IS, which might be a possible mechanism by which plasma CGRP is elevated [12].

It should be noted that numerous studies have confirmed the anti-hyperalgesia effect of EA in recurrent migraine-like rats. These studies, however, have paid more attention to the role of EA in the imbalances of the brainstem descending pain regulation system and serotonin receptors, such as 5HTR, until now [27,28,48], while studies on the improvement of pain hypersensitivity based on EA inhibition of microglial activation are limited.

### 4.2. EA Can Improve Pain Hypersensitivity by Inhibiting Microglial Activation and Microglial-Mediated Inflammatory Responses Following Repeated Dural Stimulation with IS

Neuroinflammation drives widespread chronic pain and, based on the potential role of neuroinflammatory signalling, non-neuronal cells, such as glia cells, need to be researched. The role of microglia and the microglial-mediated inflammatory cytokines in migraines has been preliminarily understood in recent years [11]. Microglia are widely distributed in the central nervous system and play a critical role in the stability of the surrounding environment. Normally, the microglial cell body is small, dense, and branched, while activation of microglia in the damaged environment or stimulation is characterised by the increased expression of microglial markers, such as Ibal-1 and CD11b, and morphological changes, including hypertrophy or process retraction and extension of microglia [53,54]. Previous evidence of microglial activation was detected following repeated dural inflammatory stimulation and widespread activation throughout the TNC during the chronic stage of migraine [12]. Other previous studies also confirmed that microglial activation contributed to pain hypersensitivity in chronic migraines [6,7,40]. In agreement with these previous studies, the results from the current study showed positive expression increases, and morphological proliferation was featured by cell body hypertrophy in microglia cells within the TNC after repeated dural stimulation with IS, supporting the notion that the activation of microglia involves the pathophysiology of a migraine and its chronic progression.

Activated microglia have a strong phagocytic capacity and can produce a variety of pro-inflammatory factors, including IL-1β, TNF-α, and IL-6 [55], and these increases of microglial-mediated cytokines exert neurogenic neuroinflammation in chronic migraines [9]. Previous studies reported that allodynia created by supradural IS increased the microglial activation marker and gene expression of proinflammatory mediators [56], and the relief of pain hypersensitivity in rats was typically associated with microglia-derived inflammatory cytokines (IL-1β, TNF-α and IL-6) [10], which means that the hyperalgesia state can be improved by inhibiting microglia and their mediated inflammation. It is well known that stimulation of the dura mater can cause elevated neuroinflammatory factors, which indicate the existence of an inflammatory state [57]. Moreover, multiple inflammatory cytokines, such as IL-1β, TNF-α, and IL-6 were significantly increased in the serum of patients with migraine attacks, showing that migraine is accompanied by systemic inflammation [41,58,59,60], which might be related to the enhanced BBB permeability in TNC induction following repeated dural stimulation [12]. According to the theory of microglia–neuron communication [6,7], microglia need to communicate with neurons to trigger their hypersensitivity, and the inflammatory factor receptors on neurons provide evidence for the inflammatory mediators secreted by microglia; therefore, we investigated the expressions of the above inflammatory cytokines. A recent in vivo study induced a rat migraine model with dural electric stimulation and found an increase in serum inflammatory factors, including IL-1β, TNF-α and IL-6 [30]. In this study, the changes in IL-1β, TNF-α, and IL-6 in the plasma were measured and the changes in the above factors in the TNC were analysed, with the aim to more fully evaluate the peripheral and central inflammatory responses. Consistently with a previous study [30], it was found that elevated levels of IL-1β, TNF-α, and IL-6 in both plasma and the TNC, reflected a persistent and systemic inflammatory state.

There have been reports that EA could inhibit microglial activation and subsequent inflammatory responses involved in neurologic disorders. A previous study has demonstrated that EA treatment promoted the degradation of β-amyloid plaques, and that this reduced microglial activation in the brains of Alzheimer’s disease (AD) model mice. Another study has found that EA tended to improve the microglia polarization in the M2 phenotype and enhance the anti-inflammatory ability in AD rat models [61]. A recent study provided the similar opinion that EA could inhibit the activation of microglia to improve memory loss in AD, while improving the M2 microglia polarization to support neuroprotection [62]. These results indicated that EA could exert effects by inhibiting microglial activation and limiting the inflammatory response. In this study, it was found that EA treatment can significantly inhibit the activation of microglia that corresponded with the improvement of hyperalgesia. This finding provided some evidence for this study’s hypothesis that EA could ameliorate hyperalgesia associated with the chronic progression of migraines by inhibiting microglial activation. Moreover, another recent study confirmed that the inflammatory cytokines, including IL-1β, TNF-α, and IL-6, significant decreased after EA treatment at GB20 and GB34 in a migraine model [30]. In line with the previous study, the evidence from this study suggests that the inflammatory cytokines were significantly reduced in the TNC and plasma after EA treatment. Thus, it was reasonable to believe that EA exerted anti-migraine effects by inhibiting the activation of microglia and subsequent inflammatory responses in a migraine rat model of IS dural stimulation.

### 4.3. EA Inhibits Microglial Activation, and Microglial-Mediated Inflammatory Responses Might Be Related to TLR4/NF-κB Following Repeated Dural Stimulation with IS

Microglia express multiple receptors. One of these receptors, TLR4, is an important trigger for inducing the activation of microglia, which could result in an inflammatory cascade reaction [15,63]. NF-κB, is a downstream molecule of TLR4 and a master regulator of inflammatory cytokines. Its activation results in the enhanced transcription and release of IL-1β, TNF-α, and IL-6 [14,64]. Specifically, TLR4 can promote NF-κB, especially the p65 subunits, to enter the cell nucleus and bind with the gene to initiate transcription, which further promotes IL-1β and TNF-α, and the synthesis of various inflammatory cytokines, ultimately promoting the formation and development of the inflammatory response [65]. These molecules also promote the activation of glial cells and the release of inflammatory cytokines, resulting in hyperalgesia via the nociceptive pathway [16,66]. Previous studies have confirmed that the activation of the TLR4/NF–κB signalling pathway induces hyperalgesia in several animal models of inflammation or neuropathic pain [67,68]. A recent study showed that activation of TLR4/NF-κB also promotes migraine-related allodynia [69]. Another study confirmed that the activation of the TLR4/NF–κB signalling pathway in the trigeminal system contributes to the development of hyperalgesia in a migraine rat model, and that the activation of microglia induced by dural inflammation is regulated by TLR4 [15]. Moreover, reduced IL-1β and TNF-α release in microglia is associated with significant inhibition of NF-κB nuclear translocation, suggesting that NF-κB is also involved in microglial inflammatory cytokine production as a critical regulator [43]. In agreement with these studies [15,69], the findings from this study indicated that repeated dural IS stimulation induced a strong upregulation of TLR4 in the TNC, and the same elevated expression of NF-κB was observed compared with only low expression in other control rats; thus, the increased expressions of TLR4/NF- κB in activated TNC microglia may be related to chronic migraines, and TLR4/NF-κB may be regulators of microglial activation and microglial-mediated inflammatory release. Furthermore, previous research observed that acupuncture could alleviate inflammatory reactions by suppressing the TLR4/NF-κB signalling pathway in microglia, and that TLR4 antagonists could mimic the anti-inflammatory effect [70,71]. Another study reported that TLR4 antagonists can inhibit microglial activation and the consequent releases of inflammatory mediators, and, similarly, block the development of allodynia after dural IS in migraine rats [56]. Consistently with these studies, it was observed that the expression of TLR4/NF-κB was significantly decreased in the TNC in repeated IS-induced migraine-like models after EA treatment, which is consistent with the inhibition of microglia, and demonstrates that EA could improve the activation status associated with microglia by reducing TLR4/NF-κB.

In addition, based on the common acupoints of clinical treatment and migraine research, GB20 was selected to exert its therapeutic effects for migraine in the adjacent area and GB34 was chosen to play a synergistic effect at a distance. Acupuncture can distantly regulate the physiology of organs via the stimulation of specific parts of the body (acupoints) [72]. Recent studies have confirmed a crucial role for autonomic nerve activity in mediating the anti-inflammatory signalling of EA. They have provided a neuroanatomical basis for the selectivity and specificity of acupoints in driving specific autonomic pathways [73], and indicate that EA can be an alternative for vagal stimulation to exert effects [72]. This may provide an explanation for the ability of EA treatment at GB20 and GB34 to enhance the anti-migraine effects in the current study. We provided evidence that EA was involved in anti-hyperalgesia of migraine by inhibiting microglia-mediated inflammatory responses through the stimulation of specific acupoints. The underlying neural mechanism of EA treatment driving acupoints needs to be further explored in the subsequent study.

## 5. Conclusions

In conclusion, in the migraine rat model, following repetitive dural stimulation with IS, it was observed that pain hypersensitivity was consistent with an increase in neuronal biomarker CGRP level. At the same time, microglial activation and increased microglial-mediated inflammatory cytokines, including IL-1β, TNF-α, and IL-6, resulting in neuroinflammation was detected. Moreover, the activation of microglia and microglial-mediated inflammatory response was regulated by TLR4 and NF-κB, which together enhanced and amplified the pain response. Importantly, the improvement of hyperalgesia, the reduction of microglial activation, and the normalisation of the above relevant inflammatory factors corresponded with behavioural improvement with EA treatment, which indicates that EA could ameliorate pain hypersensitivity in migraine-like models via microglia and microglial-mediated neuroinflammatory response to a certain extent.

## Figures and Tables

**Figure 1 brainsci-13-00541-f001:**
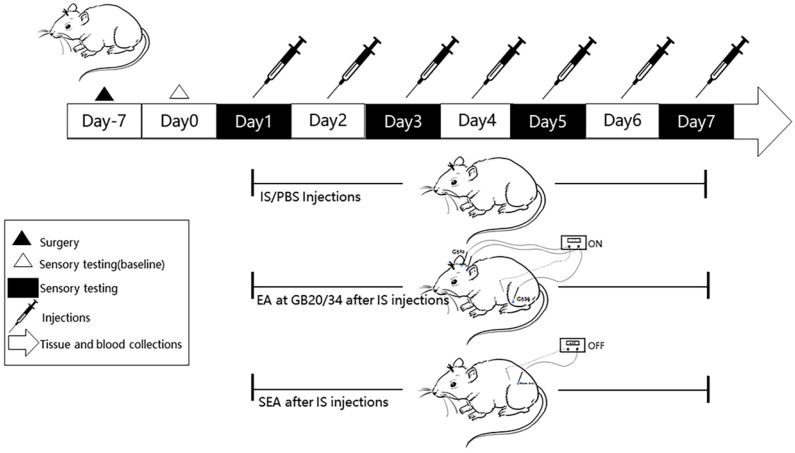
A flow chart of the experimental design. After 7 days of recovery following the implantation of the cannula onto the dura, tactile sensory testing by von-Frey filaments was performed to measure the facial mechanical withdrawal thresholds on day 0 (baseline). Rats received daily repeated-dural IS/PBS (10 ul) injections from day 1 to day 7 for a total of 7 sessions. We set up a blank group without special intervention, except that they were grasped and fixed at the same time (not shown in Figure 1). EA at GB20/GB34 was performed every day after daily IS-dural injection from day 1 to day 7, and SEAs was performed at sham-acupoints without current stimulation at the same time. Facial mechanical threshold was measured at four time points (day 1, day 3, day 5, and day 7) to assess the changes of pain hypersensitivity. The TNC Tissues and blood sample collections were performed after the last intervention.

**Figure 2 brainsci-13-00541-f002:**
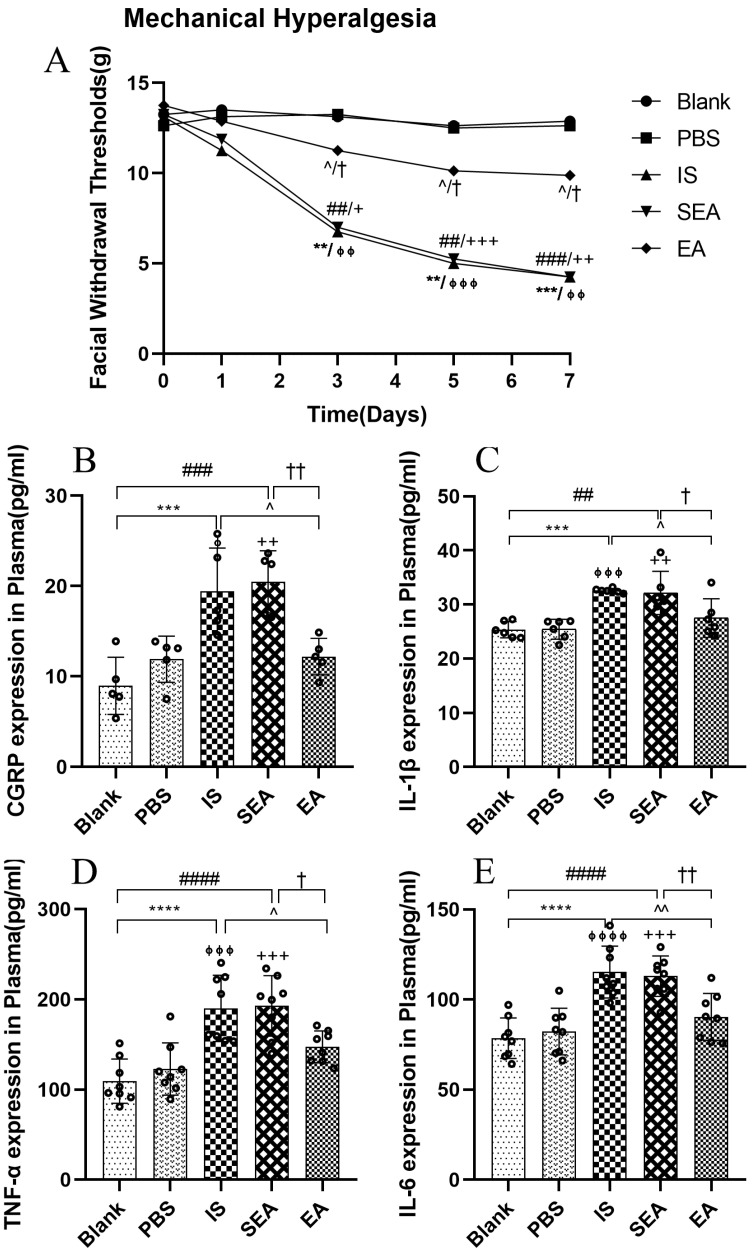
(**A**) The changes in facial mechanical withdrawal thresholds in the five groups of rats (*n* = 8/group). Repeated daily IS-dural administrations induced basal mechanical response decreases and EA treatment at GB20/34 alleviated the facial mechanical withdrawal thresholds in rats following dural IS stimulation. Plasma levels of CGRP (**B**) and inflammatory cytokines, including IL-1β (**C**), TNF-α (**D**), and IL-6 (**E**), in the five groups of rats detected by ELISA (*n* = 6–8/group). Repeated IS infusions into dura increased the release of CGRP and the above inflammatory cytokines, while EA decreased plasma levels of CGRP and inflammatory cytokines. Kruskal–Wallis test was used to analyse facial mechanical withdrawal thresholds followed by Dunn’s multiple comparisons. ANOVA followed by Tukey’s post hoc analysis test were used to analyse ELISA data. Data are presented as the mean ± standard deviation. ** *p* < 0.01, *** *p* < 0.001, **** *p* < 0.0001, Blank vs. IS; ⏀ *p* < 0.05, ⏀⏀ *p* < 0.01, ⏀⏀⏀ *p* < 0.001, ⏀⏀⏀⏀ *p* < 0.0001, PBS vs. IS; ## *p* < 0.01, ### *p* < 0.001, #### *p* < 0.0001, Blank vs. SEA; + *p* < 0.05, ++ *p* < 0.01, +++ *p* < 0.001, PBS vs. SEA; ^ *p* < 0.05, ^^ *p* < 0.01, IS vs. EA; † *p* < 0.05, †† *p* < 0.01, SEA vs. EA.

**Figure 3 brainsci-13-00541-f003:**
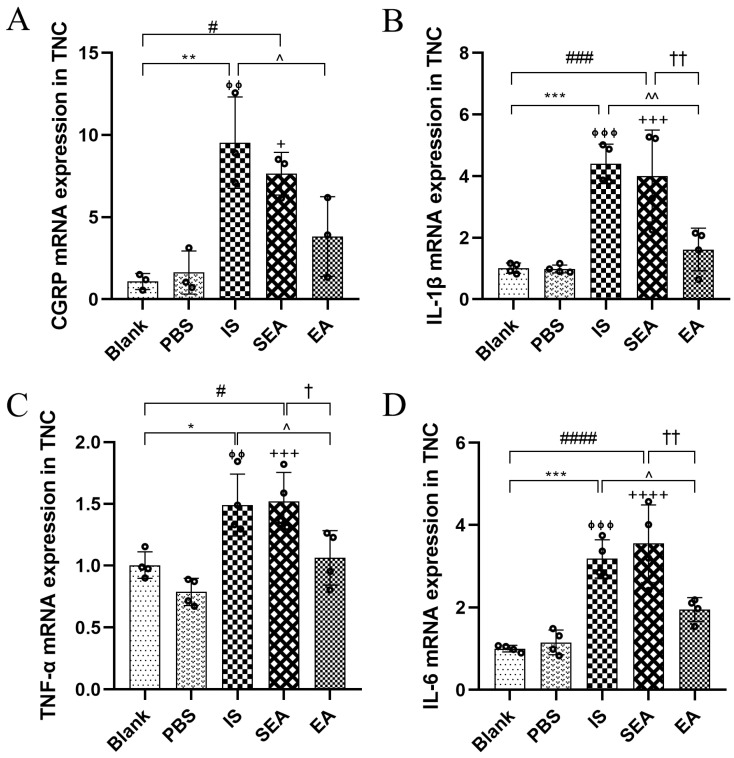
mRNA expressions of CGRP (**A**) and inflammatory cytokines, including IL-1β (**B**) TNF-α (**C**), and IL-6 (**D**), in TNC in the five groups of rats as detected by RT-PCR (*n* = 3–4/group). Repeated IS infusions into dura increased the mRNA expressions of CGRP and the above inflammatory cytokines, while EA decreased mRNA levels of CGRP and the above inflammatory cytokines. ANOVA and Tukey’s post hoc analysis test were performed and data are presented as the mean ± standard deviation. * *p* < 0.05, ** *p* < 0.01, *** *p* < 0.001, Blank vs. IS; ⏀⏀ *p* < 0.01, ⏀⏀⏀ *p* < 0.001, PBS vs. IS; # *p* < 0.05, ### *p* < 0.001, #### *p* < 0.0001, Blank vs. SEA; + *p* < 0.05, +++ *p* < 0.001, ++++ *p* < 0.0001, PBS vs. SEA; ^ *p* < 0.05, ^^ *p* < 0.01, IS vs. EA; *p* = 0.1613 (CGRP), † *p* < 0.05, †† *p* < 0.01, SEA vs. EA.

**Figure 4 brainsci-13-00541-f004:**
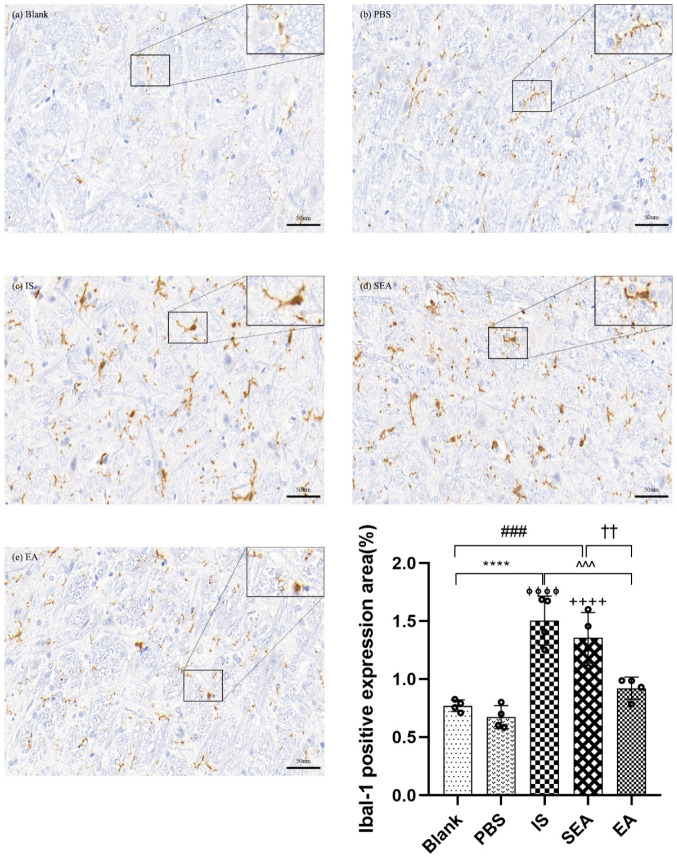
The positive expression of Iba-1-labeled microglia in TNC in five groups of rats ((**a**): Blank; (**b**): PBS; (**c**): IS; (**d**): SEA; (**e**): EA) and the percentage of positively expressed area in the total observed area (area%) (*n* = 4/group). Images of microglia labelled with Iba-1 IHC staining in TNC (400×, Scale bar = 50 µm, three sections/rat). Repeated IS infusions into dura induced the microglial activation (morphological hypertrophy and increase in positive cell expression), whereas EA reduced the activation of microglia. ANOVA and Tukey’s post hoc analysis test were performed and data are presented as the mean ± standard deviation. **** *p* < 0.0001, Blank vs. IS; ⏀⏀⏀⏀ *p* < 0.0001, PBS vs. IS; ### *p* < 0.001, Blank vs. SEA; ++++ *p* < 0.0001, PBS vs. SEA; ^^^ *p* < 0.001, IS vs. EA; †† *p* < 0.01, SEA vs. EA.

**Figure 5 brainsci-13-00541-f005:**
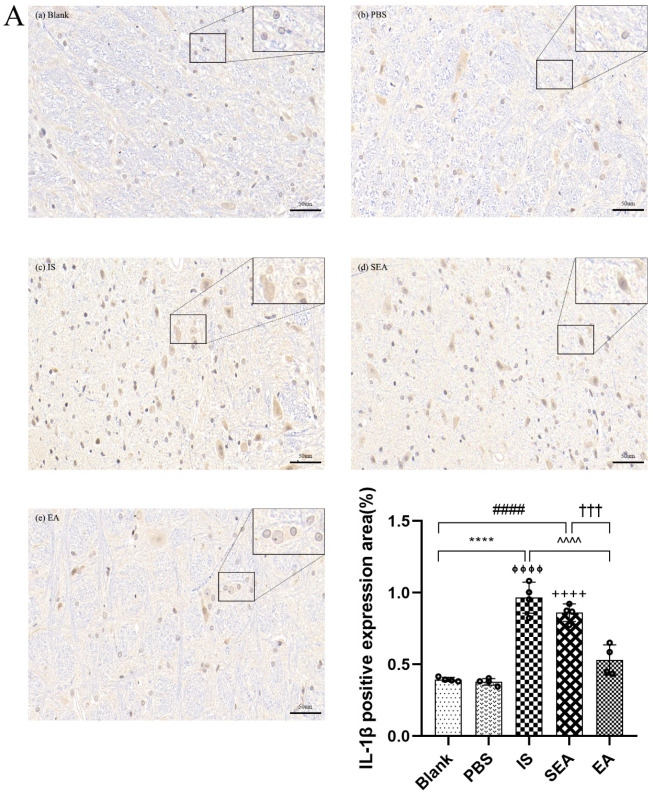
The positive expressions of inflammatory cytokines, including IL-1β (**A**), TNF-α (**B**), and IL-6 (**C**), in TNC in five groups of rats ((**a**): Blank; (**b**): PBS; (**c**): IS; (**d**): SEA; (**e**): EA) and the percentage of positively expressed area in the total observed area (area%) (*n* = 4/group). Images of inflammatory cytokines’ IHC staining in TNC (400×, Scale bar = 50 µm, three sections/rat). Repeated IS infusions into dura induced an increased number of positive cells, while EA reduced positive expressions of those inflammatory cytokines. ANOVA and Tukey’s post hoc analysis test were performed and data are presented as the mean ± standard deviation. **** *p* < 0.0001, Blank vs. IS; ⏀⏀⏀⏀ *p* < 0.0001, PBS vs. IS; #### *p* < 0.0001, Blank vs. SEA; ++++ *p* < 0.0001, PBS vs. SEA; ^^^ *p* < 0.001, ^^^^ *p* < 0.0001, IS vs. EA; ††† *p* < 0.001, SEA vs. EA.

**Figure 6 brainsci-13-00541-f006:**
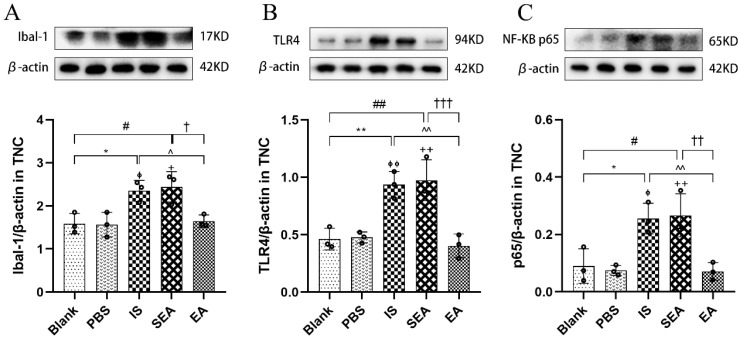
Ibal-1, TLR4, and NF-κB protein expression levels in TNC assessed by Western blot assay in five groups of rats (*n* = 3/group). Repeated IS infusions into dura induced an increased protein expression of Ibal-1, TLR4, and NF-κB, while EA decreased protein expression of the above factors. (**A**) Ibal-1 protein expression levels. (**B**) TLR4 protein expression levels. (**C**) NF-κB protein expression levels. Protein expressions levels were normalised against β-actin and analysed using ANOVA and Tukey’s post hoc analysis test, and densitometry results are shown as the mean ± standard deviation. * *p* < 0.05, ** *p* < 0.01, Blank vs. IS; ⏀ *p* < 0.05, ⏀⏀ *p* < 0.01, PBS vs. IS; # *p* < 0.05, ## *p* < 0.01, Blank vs. SEA; + *p* < 0.05, ++ *p* < 0.01, PBS vs. SEA; ^ *p* < 0.05, ^^ *p* < 0.01, IS vs. EA; † *p* < 0.05, †† *p* < 0.01, ††† *p* < 0.001, SEA vs. EA.

**Table 1 brainsci-13-00541-t001:** Sequences and Product Length of Primers used.

Gene	Primer Sequences (5′-3′)	Product (bp)
CGRP	Forward TTCCTGGTTGTCAGCATCTTG	121
	Reverse GTAGGCGAGCTTCTTCTTCACT	
IL-1β	Forward TCCCAAACAATACCCAAAGAAG	171
	Reverse ACTATGTCCCGACCATTGCTG	
TNF-α	Forward CTTCTCATTCCTGCTCGTGG	200
	Reverse CCGCTTGGTGGTTTGCTAC	
IL-6	Forward GACAAAGCCAGAGTCATTCAGAG	163
	Reverse GGATGGTCTTGGTCCTTAGCC	
β-actin	Forward ACGGTCAGGTCATCACTATCG	155
	Reverse GGCATAGAGGTCTTTACGGATG	

**Table 2 brainsci-13-00541-t002:** Antibodies used in immunohistochemistry and Western blot analysis.

Antibody	Manufacturer	Catalogue Number	Host	Dilution
For immunohistochemistry				
Ibal-1	Abcam, Cambridge, UK	ab178847	Rabbit	1:200
IL-1β	Abcam, Cambridge, UK	ab254360	Rabbit	1:50
TNF-α	Bioss, Beijing, China	bs-2081R	Rabbit	1:100
IL-6	Bioss, Beijing, China	bs-6309R	Rabbit	1:100
Anti-Rabbit IgG (HRP)	AiFang, Changsha, China	AFIHC003	Goat	1:200
For Western blot analysis				
Ibal-1	Abcam, Cambridge, UK	ab178847	Rabbit	1:1000
TLR4	Novus, St Charles, MO, USA	NB100-56566	Mouse	1:1000
NF-κB	CST, Danvers, MA, USA	6956T	Mouse	1:1000
β-actin	Zenbio, Chengdu, China	200068-8F10	Mouse	1:10,000
Anti-Rabbit IgG (HRP)	Zenbio, Chengdu, China	511203	Goat	1:10,000
Anti-Mouse IgG (HRP)	Zenbio, Chengdu, China	511103	Goat	1:10,000

## Data Availability

The data presented in this study are available upon reasonable request from the corresponding author.

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
