# Peer review of "Electroacupuncture at Fengchi(GB20) and Yanglingquan(GB34) Ameliorates Paralgesia through Microglia-Mediated Neuroinflammation in a Rat Model of Migraine"

_brainsci, 2023, doi:10.3390/brainsci13040541_

Round 1

Reviewer 1 Report

The authors' purpose in this work was to establish that electroacupuncture at points GB20/GB34 relieves pain and alleviates neuroinflammation in a rat model of migraine via the downregulation of inflammatory cytokines in microglia, as well as the regulation of microglia activation by the molecule TLR4 and the downstream molecule NF-κB.

Title Suggestion: Electroacupuncture ameliorates paralgesia through microglia-mediated neuroinflammation in a rat model of migraine

Results:

Typos in Figure 2B and 2D, 3A and 3B, 4, 5A, 5B, 5C (graphics): exprssion

Figure 6. Improve the quality of figure 6 (Western blot). Remove the word "density" from the y-axis. 

The limitations of the study need to be reconsidered as they are not consistent with what was proposed in the study.

Reviewer 2 Report

The subject and the experimental model is quite interesting and I consider it to have important relevance, however I list the following suggestions and doubts:

The introduction is quite descriptive and there is information that is repeated in the development of the results and discussion, the author focuses on discussing more regarding the reproduction of the experimental model and subtracts an explanation of why the anti-inflammatory effect of AD is that the trigeminal pathway generates inactivation of microglia and subsequent inhibition of proinflammatory cytokines. In addition to the fact that the stimulation of GB20 could be related to the trigeminal pathway, but GB34 due to its location, in what way would it be involved? Since studies on the anti-inflammatory effect of vagus nerve stimulation through certain formulations of EA acupuncture points have recently been reported, I consider it would be interesting to delve further into this regard.

In relation to the evaluation of pain, does this analgesic effect last after the time of AE?

On each day of treatment, an evaluation was carried out before and after the IS and in the same way before and after the treatment with EA and SHAM?

I think it would be prudent to compare the effect of EA with a drug control group and not only with a sham group without stimulation, as well as to observe the expression of Iba1 and its colocalization with IL1b, TNF-a and IL-6 by immunofluorescence, and to evaluate the differences by analysis of intensity of fluorescence.

Reviewer 3 Report

The review would like to declare no potential conflict of interest with the authors nor the affiliated institutions.

Overall, Zhou and team presented an interested article describing the possible application of electroacupuncture (EA) on migraine, using an animal model mimicking migraine-like responses. However, several major concerns must be addressed. 

1. The title does not fully reflect the content of the study. Suggest to specify which acupuncture points (acupoints) are being studied in the manuscript. The current title are too general which may mislead the readers that are not familiar with EA that all acupoints will have the similar therapeutic effect.

2. "Drugs/Chemical" sub-section is missing in the material and methods section. The volume of administration for i.p. treatment is missing. Furthermore, the content of inflammatory soup is not specified. 

3. As intra-dural cannulation is employed in this study, the authors did not specify or describe what measures have been taken to ensure that the cannulation was done accurately? Any methods of detection of misplanting the cannulae? 

4. In tactile sensory testing, it was stated that the rats may receive multilple (up to 5) times of filament "pricking". Kindly specify the interval duration in between each stimulation.

5. The statistical anaylsis subsection is kindly messy. The font and spacing format is inconsistent. The post hoc analysis was not specified. The number of animal used per group should be specified. 

6. It is unclear if the n number used for each test. The bar graph showed inconsistent n number. some bar graph has up to 6 points, whereas some only have n=4. Kindly explain.

7. The results description is very messy, especially the description of statistical difference. It is not necessary to introduce multiple type of symbols in the results and figure legend. It may confuse the readers. again, the post hoc analysis was not specified.

8. The authors did not explain how they quantify the staining intensity. Furthermore, how does the staining intensity being used to compare among different treatment groups?

Round 2

Reviewer 3 Report

The concerns were addressed.

however, language editing is still required before final publicaiton.